# Neurofibromatosis Type 2 (NF2) and the Implications for Vestibular Schwannoma and Meningioma Pathogenesis

**DOI:** 10.3390/ijms22020690

**Published:** 2021-01-12

**Authors:** Suha Bachir, Sanjit Shah, Scott Shapiro, Abigail Koehler, Abdelkader Mahammedi, Ravi N. Samy, Mario Zuccarello, Elizabeth Schorry, Soma Sengupta

**Affiliations:** 1Department of Genetics, Cincinnati Children’s Hospital, Cincinnati, OH 45229, USA; suha.bachir@cchmc.org (S.B.); elizabeth.schorry@cchmc.org (E.S.); 2Department of Neurosurgery, University of Cincinnati, Cincinnati, OH 45267, USA; Sanjit.Shah@UCHealth.com (S.S.); Mario.Zuccarello@UCHealth.com (M.Z.); 3Department of Otolaryngology, University of Cincinnati, Cincinnati, OH 45267, USA; scott.shapiro@cchmc.org (S.S.); Ravi.Samy@UCHealth.com (R.N.S.); 4Department of Neurology, University of Cincinnati, Cincinnati, OH 45267, USA; koehleai@ucmail.uc.edu; 5Department of Radiology, University of Cincinnati, Cincinnati, OH 45267, USA; mahammar@ucmail.uc.edu

**Keywords:** neurofibromatosis type 2 (NF2), meningiomas, vestibular schwannomas

## Abstract

Patients diagnosed with neurofibromatosis type 2 (NF2) are extremely likely to develop meningiomas, in addition to vestibular schwannomas. Meningiomas are a common primary brain tumor; many NF2 patients suffer from multiple meningiomas. In NF2, patients have mutations in the *NF2* gene, specifically with loss of function in a tumor-suppressor protein that has a number of synonymous names, including: Merlin, Neurofibromin 2, and schwannomin. Merlin is a 70 kDa protein that has 10 different isoforms. The Hippo Tumor Suppressor pathway is regulated upstream by Merlin. This pathway is critical in regulating cell proliferation and apoptosis, characteristics that are important for tumor progression. Mutations of the NF2 gene are strongly associated with NF2 diagnosis, leading to benign proliferative conditions such as vestibular schwannomas and meningiomas. Unfortunately, even though these tumors are benign, they are associated with significant morbidity and the potential for early mortality. In this review, we aim to encompass meningiomas and vestibular schwannomas as they pertain to NF2 by assessing molecular genetics, common tumor types, and tumor pathogenesis.

## 1. Neurofibromatosis Type 2 (NF2): Introduction and Genetic Overview

Neurofibromatosis type 2 (NF2) is an autosomal dominant condition caused by pathogenic variants in the *NF2* gene (*NF2*; MIM # 607379) causing loss of function of the tumor suppressor protein, Merlin [1,2,3]. NF2 is characterized by central and peripheral nervous system (CNS and PNS) tumors [3].

The incidence of NF2 is around one in 25,000 with a penetrance of 95% [4]. Over half of patients with NF2 are reported to have a de novo mutation and around one-third are mosaic [1]. Symptom onset is usually by age 20 with approximately 90% of patients having the pathognomonic feature of the disease: bilateral vestibular schwannomas. Around 50% of patients also have meningiomas [4]. Other common tumors include spinal schwannomas, ependymomas, and dermal schwannomas [5].

Clinical diagnostic criteria for NF2 have evolved over time, and a new revision is due to be published in the near future. The clinical diagnostic criteria published in 2017 are detailed below in Table 1 [6], and tumor types are summarized in Table 1. Interestingly, NF2 is genetically unrelated to the more common neurofibromatosis type 1 (NF1) which is due to pathogenic variants of the tumor suppressor *NF1* gene on chromosome 17. It is, however, closely related to schwannomatosis, due to variants in either *INI1 (SMARCB1)* or *LZTR1*, which are closely located to *NF2* on chromosome 22. As the predominant tumor type in NF2 is the schwannoma (and not neurofibroma), there has been discussion that the more appropriate name for NF2 might be Schwannomatosis Predisposition Syndrome (SPS), merlin type.

Treatment of NF2 involves the combination of medical surveillance through physical exam, audiometric testing, imaging, and surgical intervention when indicated. Patients are managed by a multidisciplinary team including neurotologists, neurologists, audiologists, oncologists, geneticists, neurosurgeons, and ophthalmologists [7].

## 2. NF2; Molecular Genetics

*NF2* is a tumor suppressor gene comprised of 17 exons with 2 splicing isoforms that is positioned on chromosome 22q12.2. It encodes the 595 amino acid protein, Merlin [3]. Merlin is a member of the Ezrin/Radixin/Moesin (ERM) family of membrane–cytoskeleton-linking proteins with an enigmatic role, although there is evidence to suggest it is involved in stabilizing the membrane cytoskeleton interface by inhibiting signals involving PI3kinase/Akt, Raf/MEK/ERK, and mTOR signaling pathways [7,9,10]. The mechanism of tumorigenesis in NF2 has yet to be fully elucidated, although loss of heterozygosity involving allelic loss of *NF2* is thought to be a likely mode, as evidenced by work on skin tumors, vestibular schwannomas, and meningiomas in NF2 patients [11,12]. Other data have suggested an epigenetic role involving transcriptional inactivation of the *NF2* gene from hyper-methylation as another possible tumorigenesis mechanism [13].

Typically, NF2-affected family members experience the same type and location of the *NF2* germ-line variant, in which phenotypic expression of NF2 correlates amongst family members. Intra-familial similarity and the severity of phenotype is significant as variations in phenotype can be associated with mutations causing truncated protein expression [4]. Stochastic or epigenetic factors are certainly at play as evidenced by phenotypic variability seen in monozygotic twins [14]. Severity is categorized by early age of onset, hearing loss, and increased numbers of meningiomas [2,3]. Nonsense and missense variants are associated with a more severe and mild phenotype, respectively, whereas splice-site variants are more variable [15,16]. Incidentally, there are reports in the literature of mutations at the 5-prime end of the *NF2* gene that are associated with increased intracranial meningiomas [6]. This genotype-phenotype correlation highlights the importance of offering genetic testing. Next-generation sequencing of all 17 coding exons of the *NF2* gene is the best molecular test with up to 90% variant detection rate with a positive NF2 family history; a lower detection rate ranging between 25–60% is present in sporadic cases, likely due to somatic mosaicism [1,17]. When genetic testing is offered early to patients with suspected NF2, it provides useful prognostic information and a more tailored therapeutic approach. Selvanatham et al. performed molecular genetic analysis on 268 NF2 patients and found that those with nonsense variants had a more severe phenotype including more meningiomas and spinal tumors. In fact, they were diagnosed at an earlier age, which sadly may not result in an improved overall outcome but did allow earlier intervention [18].

## 3. NF2: Tumor Types

The tumor types associated with NF2 including incidence, clinical presentation, histological, imaging, and treatment/complications are summarized in Table 2 [4,19,20,21]. The neuroimaging hallmarks give rise to the acronym MISME, which describes multiple inherited schwannomas, meningiomas, and ependymomas [22].

## 4. NF2 Meningioma Pathogenesis

Meningiomas can be intracranial or spinal in NF2. These are common primary intracranial dural-based tumors arising from arachnoid cap cells, with an incidence of 7–8/100,000 people/year; they account for 37.1% of primary intracranial tumors [23,24,25]. The clinical presentation of meningioma ranges from incidental discovery to headaches, visual deficits, cranial nerve dysfunction, and seizures due to mass effect or cortical irritation (please see Figure 1 and Figure 2 for an example of meningiomas in a patient). According to World Health Organization (WHO) classification, 15 subtypes of meningioma exist, with grades corresponding to histopathologic analysis [26,27]. However, studies have demonstrated that WHO grading correlates poorly with prognosis as grade alone is not entirely predictive of recurrence and malignant transformation [28,29], creating a role for molecular genetics both in meningioma treatment and prognostication. An estimated 50–75% of patients with NF2 develop meningiomas; in contrast to sporadic meningiomas, these are often Grade II or III, have a worse prognosis, and higher rate of recurrence [30,31,32].

Loss of chromosome 22 has been widely implicated in the pathogenesis of meningioma, sharing a common pathway for tumorigenesis with NF2 patients due to the presence of neurofibromin on chromosome 22q12.2. Deletions, nonsense mutations, splice site mutations, and translocations in *NF2*/Merlin are identified in 50–60% of the general population of patients with meningiomas, and loss of chromosome 22 in tumor tissue can be seen in 40–80% of patients developing meningioma [23,24]. The exact mechanism by which loss of Merlin or chromosome 22 affects meningioma pathogenesis is not well understood, as both have been implicated to have a role in cytoskeletal remodeling; indeed, the production of junctional proteins E-cadherin and Zo-1 demonstrate a positive correlation with *NF2*/Merlin expression [33]. Furthermore, loss of chromosome 22 has been widely shown to activate oncogenic pathways with downstream targets such as Ras/mitogen-activated protein kinase, Notch, and mammalian target of rapamycin (mTOR) [34]. Notably, Bi et al. demonstrated that a mean of 23 *NF2* mutations in high-grade gliomas compared to 11 in low-grade gliomas, suggesting a positive correlation between *NF2* mutation rate and meningioma grade; this has been corroborated by additional studies [23,30,33]. Interestingly, a recent study by Angus and colleagues showed upregulation of erythropoietin producing hepatocellular receptor A2 (*EPH-RA2*), a downstream target *MEK*, in *NF2* null mice; therapeutic interventions targeting such upstream signaling cascades of receptor tyrosine kinases may prove efficacious in treatment of *NF2* related meningiomas [35].

In addition to the downstream effects of germline mutations to *NF2*, it is worth mentioning that a number of sporadic mutations have been implicated in the pathogenesis of meningioma. Several such mutations, including those to smoothened (*SMO*), tumor receptor-associated factor 7 (*TRAF7*), and phosphatidylinositol-4,5-bisphosphate 3 kinase catalytic subunit α (*PIK3CA*) are present in non-NF2 tumors and are mutually exclusive to mutations in *NF2* [31,36,37,38,39,40]. Other genomic mutations, such as those to the telomerase reverse transcriptase (*TERT*) promoter region, demonstrate co-occurrence with *NF2* mutations. While 6% of meningiomas possess *TERT* promoter mutations, nearly 80% of *TERT* mutations in meningioma co-occur with *NF2* mutations and are associated with higher tumor grade and significantly decreased tumor-free progression [28]. The co-occurrence of such mutations provides valuable targets for prognostic evaluation and therapeutic intervention.

Elucidating the mechanism behind meningioma pathogenesis at a genetic level in neurofibromatosis 2 has clinical implications. A study by Clark et al. noted that NF2-mutated meningiomas had a predilection for the posterior and lateral skull base, tentorium, and cerebral falx, while sporadic mutations, such as those to *TRAF7* and *SMO*, had a tendency for anterior skull base presentation [31]. However, in NF2, meningiomas may also occur in the spinal canal, optic nerve sheaths, and cerebral ventricles. Unlike sporadic mutations, *NF2* mutations are far more likely to produce multiple meningiomas and occur in a younger population; the mean age of meningioma diagnosis is 30 years. By age 70, the cumulative incidence of meningioma is 80% in patients with proven *NF2* mutations [23,30,41,42]. Understanding the genomic and molecular pathogenesis of *NF2*-mutated meningiomas will provide insight into meningioma tumorigenesis as a whole and offer new avenues for efficacious therapies.

## 5. NF2 Vestibular Schwannoma Pathogenesis

Vestibular schwannomas (VS) are tumors derived from the Schwann cells of the vestibular branches of the vestibulocochlear nerve (cranial nerve VIII). The most common symptoms of VS are hearing loss, tinnitus, imbalance, facial numbness/paraesthesias, and facial paresis. Although benign histologically, they may compress the brainstem leading to hydrocephalus and neurologic emergency. Rarely, tumors may also cause hydrocephalus through poorly understood mechanisms that may involve high protein levels in the subarachnoid space [43]. Unilateral, sporadic tumors account for 95% of all VS, with bilateral VS being diagnostic of NF2 [44]. When possible, sporadic and NF-2 related VS should be distinguished from schwannomatosis, which is a distinct clinical and molecular entity which results in schwannomas throughout the body. Unlike NF2, it rarely involves the vestibular nerves. This can be difficult to distinguish in patients who present initially with non-vestibular schwannomas suggestive of shcwannomatosis and later develop bilateral VS, establishing the diagnosis of NF2. It may also be difficult in patients with non-vestibular schwannomas who are mosaic for NF2 [45].

Molecular studies have shown that VS develop as a result of inactivation of both alleles of the *NF2* tumor suppressor gene. Consistent with the autosomal dominant inheritance pattern, patients with NF2 inherit one mutated copy of the *NF2* gene, and loss of heterozygosity (and all tumor suppressor function) in the Schwann cells allows for VS development [46,47,48,49,50,51,52]. NF2 is highly penetrant; patients who inherit an abnormal *NF2* gene have a 95% chance of developing bilateral VS. Approximately 50% of NF2 patients have no family history of NF2, suggesting a significant proportion of cases arise due to new germ-line mutations, rather than through inheritance, and 33% of these patients exhibit mosaicism, where the mutation takes place after conception and results in two separate cell lineages which may make mutations difficult to detect in peripheral blood analysis [53,54]. Tumor sequencing with matched peripheral blood analysis in patients with unilateral (sporadic) VS, demonstrate biallelic mutations in tumor specimens, which are not present in peripheral blood consistent with a “two-hit” theory for sporadic tumor development, which is consistent with the later presentation and less aggressive course of patients with sporadic tumors compared to their NF2 counterparts (please see Figure 3 for an example of NF2 VS) [55].

The *NF2* gene is located on chromosome 22 at 22q12.2, and codes for the protein Merlin, which has also been called schwannomin [47,48]. Alterations in *NF2* genes have been found in most but not all VS in both sporadic and syndromic-associated tumors. Interestingly, Merlin activity is nevertheless altered in cases where mutations are not detected, suggesting other post-translational events that can affect Merlin activity, or that some mutations are not being detected. The types of *NF2* gene mutation has been loosely associated with mortality and disease severity in both NF2 associated and sporadic tumors [55,56]. For NF2 patients, nonsense or frameshift mutations which result in truncated protein products generally have more severe phenotype (earlier presentation and more aggressive tumors) than missense mutations [56], though not all studies have found such an association [47].

Though well-characterized as a tumor suppressor gene, the mechanisms behind Merlin’s tumor suppressor functions have not been fully elucidated. Overexpression of Merlin in mouse models limits cell growth and cell transformation by the *ras* oncogene; growth control is lost when *NF2* is inactivated in Schwann and meningeal cells. Merlin is structurally similar to proteins of the erythrocyte 4.1-related superfamily. These proteins link the actin cytoskeleton to the cell membrane, and Schwann cells from NF2 tumors show significant alteration in the actin cytoskeletal organization. Merlin is also known to localize to the cell membrane-cytoskeletal interface. It requires a dephosphorylated state to assume its active form; this phosphorylation status is dependent on the upstream activity of cyclic-AMP dependent protein kinase A, p21-activated kinases, and myosin phosphotate-1 protein activity. At the cell membrane-cytoskeletal interface, Merlin has been shown to interact with various cytosolic proteins, cell membrane components, cell-to-cell adhesion proteins, and cytoskeletal components. In addition to its actin-cytoskeletal effects, evidence exists for a role in intracellular pro-mitotic/anti-mitotic processes. Merlin has been shown to affect various downstream mitogenic signaling pathways such as the phosphoinositide-3 kinase (PI3K) and the mitogen-activated protein kinase (MAPK) signaling pathways. These pathways are known to be involved in oncogenesis, as well as being critical for cell growth and proliferation [57,58,59,60,61,62,63,64].

## 6. The Current State and Future Directions for NF2 Related Meningiomas and Vestibular Schwannomas

NF2 is a complicated neuro-cutaneous genetic disorder that can cause considerable morbidity (Figure 4). Apart from the vestibular schwannomas and meningiomas, ependymomas and spinal schwannomas can also occur in this patient population, however, this review is focusing on the first two diseases. More treatment options need to be available, as the current options are not curative. Surgical resection can have recurrence, and the current therapeutic options such as Bevacizumab or mTOR inhibitors do not produce a long-lasting effect in these patients. As discussed in the prior sections, as the molecular pathology is being unraveled, we will be able to design better drugs for the tumor types in NF2 patients.

The molecular pathogenesis of NF2 is complicated by the fact that the loss of function of Merlin, a tumor suppressor protein that interacts with a whole host of integral signaling pathways including Hippo, receptor tyrosine kinase, Ras, MAP kinase (mitogen-activated protein kinase), p21-activated kinases, PI_3_ Kinase, Focal Adhesion Kinase, CD44, Rac/Rho pathway for cell motility, c-Jun *N*-terminal kinases, and Wnt. Therefore, targeting Merlin is extremely challenging (reviewed by Coy et al. [20]). Focusing on the molecular aberrations in the multiple meningiomas found in NF2 is still in the process of being unraveled as the genetic profiling of NF2 associated meningiomas is ongoing. However, a recent study compared the molecular profiling of cranial and spinal meningiomas in NF2 [12]. Pemov et al. [12] state that 45–58% of NF2 patients have intracranial meningiomas and 20% have spinal meningiomas. They also clarify that meningiomas in NF2 patients are usually benign, WHO Grade 1, and multiple. Unlike sporadic meningiomas, they found a lack of mutations, the most obvious and apparent link is the inactivated *NF2* gene. Interestingly, there is an ongoing clinical trial (NCT02523014) that has recently closed to accrual at UCSF looking at GSK2256098 (FAK inhibitor, also a target of Merlin) in NF2 progressive meningiomas.

Meningiomatosis, reviewed by Coy et al. [20], is associated with multiple meningiomas, and genetic analyses of these suggest that they are of clonal origin. These may be rarely associated with malignant sarcomatous lesions, and some patients have been reported to have an absence of germline *NF2* or *SMARCB1* mutations. In addition, meningiomatosis can demonstrate multiple recurrent chromosomal alterations including monosomy 22, 5q loss, and 6q gain [20]. Meningiomatosis is thought to occur through the silencing of *SMARCB1* expression and additional chromosomal alterations. Drugs such as Pazopanib which acts on inhibiting intracellular tyrosine kinase of VEGFR and PDGFR would be an interesting way to treat patients with meningiomatosis, as PDGFR and FGFR pathways are dependent on *SMARCB1* loss, and this agent is currently being looked at in a cancer known as atypical teratoid rhabdoid tumor (ATRT) [65].

NF2-related vestibular schwannomas in the recurrent setting are often treated with Bevacizumab [66]. Unfortunately, this often ends up failing, and the INTUITT-NF2 trial (Innovative Trial for Understanding the Impact of Targeted Therapies in NF2, NCT04374305) under the umbrella of the Children’s Tumor Foundation and Takeda Pharmaceuticals led by Scott Plotkin (Massachusetts General Hospital) and Jaishri Blakely (Johns Hopkins University) is underway to look at patients with vestibular schwannomas, ependymomas, and meningiomas in the NF2 population. They are currently using Brigatinib, but have a basket trial design, therefore, if the patient’s tumor does not respond, they will be eligible for another agent in this trial.

Due to the complexity of care of NF2 patients, and other neurogenetic patients, often clinical centers run a multidisciplinary clinic to address the issues of these patients (Figure 4). NF2 patients tend to be managed by specialized family physicians, neurogeneticists, neuro-oncologists, neuro-otologists, neurosurgeons, neuro-ophthalmologists, neuroradiologists, social work, audiology, speech therapy, physical therapy, and occupational therapy. As more personalized medicine approaches develop, there will be even more providers involved in the clinical care of NF2 patients who have complex clinical needs. Resources and insurances will all need to align in the increasing costs of taking care of patients with complex needs. However, if there are ways to prevent the morbidities in NF2 patients in the first place, such as gene therapy, morbidity would be less. Attempts are underway to do this. Transgenic murine NF2 models are being utilized to study drugs [67,68]. Indeed, MEK 1 and 2 inhibitors are being studied for NF2-related vestibular schwannomas [68]. In addition, one might be able to leverage microsatellite instability in meningiomas [69] and changes in *MLH1* and the *MSH2* genes may pave the way of trailing immunotherapy in recurrent meningiomas [23,70]. As previously discussed in the Children’s Tumor Foundation forum, Dr. M. Wootton, CEO of NF2 Therapeutics, mentioned that they have a monkey model of NF2 gene replacement, and based on this, he went on to state that the first human clinical trials for this *NF2* transgene may start in 2022 or 2023. Therefore, the future of NF2 is evolving.

## Figures and Tables

**Figure 1 ijms-22-00690-f001:**
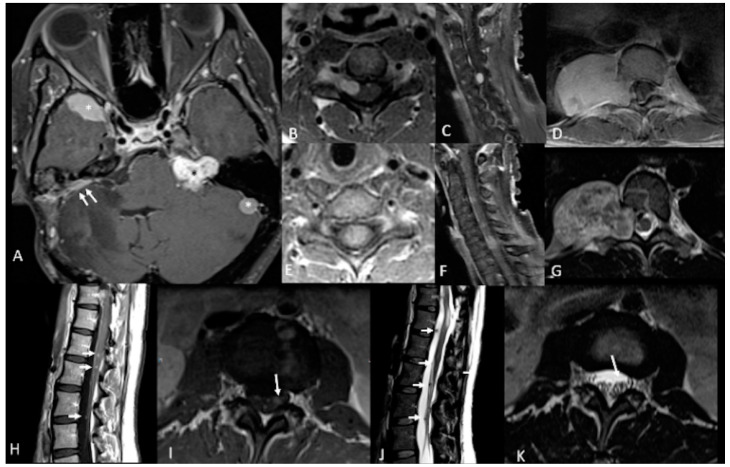
46-year-old woman with neurofibromatosis type 2 (NF2). Contrast-enhanced T1-weighted-Fat saturation MR images (**A**,**F**,**I**) show avidly enhancing schwannoma involving left cerebellopontine angle and internal auditory canal (black asterisk in (**A**)) and avidly enhancing meningiomas in the right anterior temporal convexity, sphenoid wing, and left lateral posterior fossa (white asterisks in (**A**)). Post-operative changes from right suboccipital craniotomy and mastoidectomy with residual meningioma (white arrows in (**A**)). Avidly enhancing dumbbell-shaped right C5 intradural schwannoma extending through right C5 foramen (**B**,**C**). Large avidly enhancing right paraspinal T10 schwannoma extending through right T10 foramen (**D**) with associated heterogenous T2 hyperintensity on axial T2-weighted image (**G**). Avidly enhancing intramedullary C6 ependymoma (**E**,**F**). Numerous multilevel tiny enhancing rounded nodules along the cauda equina (**H**,**I**) consistent with schwannomas which demonstrate T2 hypointensity on axial and sagittal T2-weighted images (white arrows in (**J**,**K**)).

**Figure 2 ijms-22-00690-f002:**
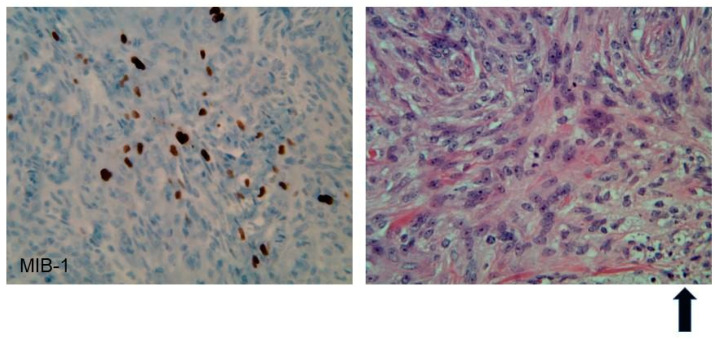
Immunohistochemistry of an atypical meningioma (WHO Grade II) from an NF2 patient. Left panel: 40× magnification, MIB-1(Ki67) nuclear immunostain shows an elevated index (25%). Right panel: 40× H&E shows increased cellularity, prominent nucleoli, mitoses, and necrosis (as represented in the bottom right hand section and highlighted by the arrow in bold).

**Figure 3 ijms-22-00690-f003:**
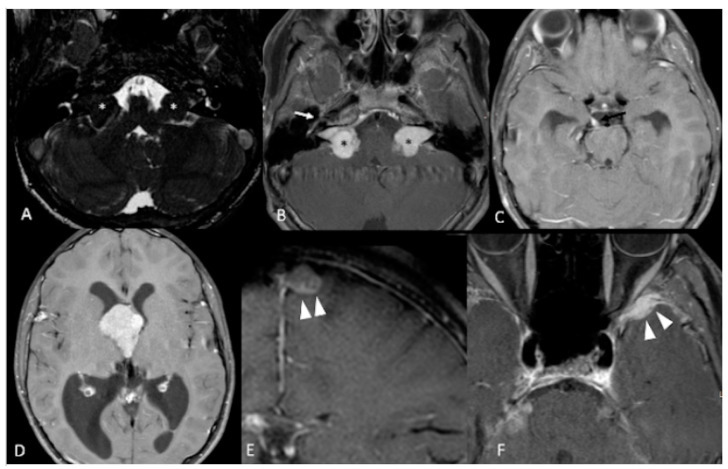
12-year-old boy with NF2. Contrast-enhanced T1-weighted-Fat saturation MR images (**B**,**F**) show avidly enhancing masses involving cerebellopontine angles and internal auditory canals (black asterisks in (**B**)) with associated T2 hypointensity on axial high resolution T2 FIESTA (white asterisks in (**A**)), consistent with bilateral vestibular schwannomas. Also noted are asymmetric enhancement and enlargement of right geniculate ganglion of right CN7 (white arrow in (**B**)) and right CN3 (black arrow in (**C**)) consistent with additional schwannomas. Large third ventricular meningioma with obstructing hydrocephalus (**D**). Additional left parafalcine and left sphenoid wing meningiomas are noted (arrowheads in (**E**,**F**)).

**Figure 4 ijms-22-00690-f004:**
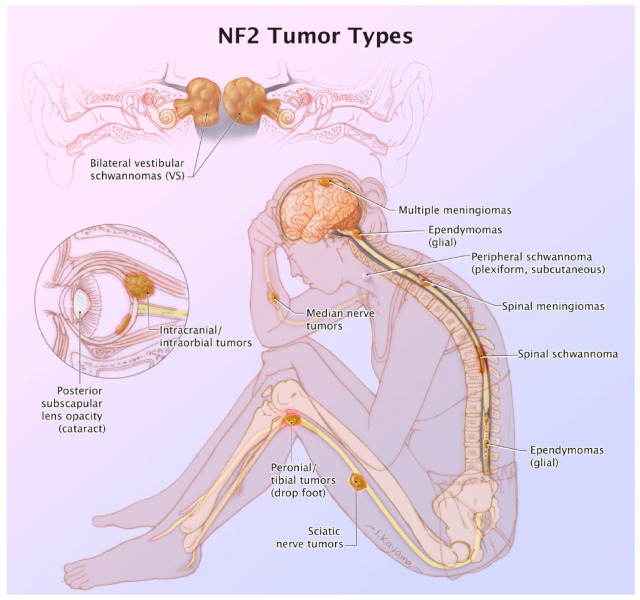
Summary figure of the various morbidities associated with NF2. An original illustration by Ikumi Kayama.

**Table 1 ijms-22-00690-t001:** Clinical diagnosis criteria for neurofibromatosis type 2 (NF2) [6,8].

1. Bilateral vestibular schwannomas < 70 years of age.
2. Unilateral vestibular schwannoma < 70 years and a first-degree relative with NF2.
3. Any two of the following: meningioma, schwannoma (non-vestibular), ependymoma, cerebral calcification, cataract *AND* first-degree relative to NF2 *OR* unilateral vestibular schwannoma and negative *LZTR1* testing. Note: recent data have excluded glioma in the criteria.
4. Multiple meningiomas and unilateral vestibular schwannoma or any two of the following: schwannoma (non-vestibular), neurofibroma, glioma, cerebral calcification, cataract.
5. Constitutional or mosaic pathogenic *NF2* gene mutation from the blood or by the identification of an identical mutation from two separate tumors in the same individual.

**Table 2 ijms-22-00690-t002:** Characteristics of NF2 tumor type.

NF2 Tumor Types	%	Clinical Presentation	Histology	Imaging	Treatment	Complications
Vestibular Schwannomas	~90%	Tinnitus Hearing loss Ataxia	Antoni A, B regions Verocay bodies Hyalinzed vessels	Often bilateral. Slightly T1 hypointense (63%) or isointense (37%). Heterogeneously T2 hyperintense (Antoni A: relatively low, Antoni B: high), cystic degenerative areas may be present if large tumor. Intense contrast enhancement on T1 C+ (Gd)	Radiosurgery Chemotherapy; Bevacizumab	Facial nerve injury Malignant transformation
Peripheral Schwannomas -Tumorlets -Plexiform	~70%	Neuropathic pain Loss of sensation Weakness Tumors on skin, head and neck region (Plexiform)	Antoni A, B regions Verocay bodies Hyalinzed vessels Infiltration of nerve	T1: 75% are isointense, 25% are hypointense. T2: more than 95% are hyperintense, often with mixed signal. Intense contrast enhancement on T1 C+ (Gd)	Intraneural dissection Excision	Rarely undergo malignant transformation although high risk of nerve infiltration
Meningiomas	~50% (20% are in kids)	Headache Seizure	Fibrous morphology Psamomma Bodies High mitotic index	Intense and homogeneous enhancement. Frequent cystic components Can be multiple Present in unusual locations: craniocervical junction.	Surgical excision Radiosurgery Current clinical trial: mTORC1/2 inhibitor AZD2014 (NCT02831257, NCT03071874)	Malignant transformation Invasion to vascular brain structures Compression effect
Ependymoma Glial	~30%	Asymptomatic	Perivascular pseudorosettes Ependymal rosettes	Usually spinal intramedullary (not intracranial/intraventricular). “String of pearls” appearance along the spinal cord and cauda equina.	Monitoring/surveillance Surgical resection if symptomatic	Malignant transformation is rare
Menigioangiomatosis	rare	Headache Seizures Behavioral changes Cortical blindness Paresis	Plaque like leptomeningeal and perivascular proliferation Fibroblastic and meningothelial appearing cells	Cortical/subcortical white matter mass characterized by Ca⁺⁺, enhancing meningovascular proliferation. Most common in temporal and frontal lobes.	Surgical excision	Intracerebral hemorrhage
Glial micro hamartomas	Common	Asymptomatic	Atypical pleomorphic nuclei, Occasional multi-nucleation, Eosinophilic cytoplasm	Cortical hyperintense T2/FLAIR lesions “Transmantle sign”	Surveillance and monitoring	None

Note: % refers to incidence.

## Data Availability

No new data were created or analyzed in this study. Data sharing is not applicable to this article.

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
