# Peer review of "Neurofibromatosis Type 2 (NF2) and the Implications for Vestibular Schwannoma and Meningioma Pathogenesis"

_ijms, 2021, doi:10.3390/ijms22020690_

Round 1
Reviewer 1 Report
The paper brings novel insight into complex NF2 syndrome. The authors focus on meningiomas and vestibular schwannomas as they pertain to NF2 by assessing molecular genetics, common tumor types, and tumor pathogenesis. The paper is well-written and concise. However, the cited literature needs to be improved. I would recommend more references. The following references should be included. A review on key molecule merlin- Pećina-Šlaus N. Merlin, the NF2 gene product. Pathol Oncol Res. 2013 Jul;19(3):365-73. doi: 10.1007/s12253-013-9644-y. Epub 2013 May 12. PMID: 23666797. https://pubmed.ncbi.nlm.nih.gov/23666797/
Additionally, more space should be given to microsatellite instability (MSI) that is part of meningioma genetic profile on page 4, line 137. Please include reference on the involvement of MSI in meningioma. Pećina-Šlaus N, Kafka A, Lechpammer M. Molecular Genetics of Intracranial Meningiomas with Emphasis on Canonical Wnt Signalling. Cancers (Basel). 2016 Jul 15;8(7):67. doi: 10.3390/cancers8070067. PMID: 27429002; PMCID: PMC4963809. https://pubmed.ncbi.nlm.nih.gov/27429002/
The figures and table are excellent although I do not understand why the arrow on figure 1 is below the figure.
Minor points: please include full name of the condition when first mentioned in the Abstract section.
In my opinion this is a valuable contribution to understanding of NF2 syndrome. All things considered I recommend that the paper is accepted if the authors are prepared to incorporate minor changes.
Author Response
I would like to personally thank reviewer's one's comments, which were extremely helpful.
1. I added Pećina-Šlaus N. Merlin, the NF2 gene product. Pathol Oncol Res. 2013 Jul;19(3):365-73. doi: 10.1007/s12253-013-9644-y. Epub 2013 May 12. PMID: 23666797. https://pubmed.ncbi.nlm.nih.gov/23666797/ and took out an older review, Yohay, K.H. The Genetic and Molecular Pathogenesis of NF1 and NF2. Semin. Pediatr. Neurol 2006, 13, 21–26. Doi: 10.1016/j.spen.2006.01.007.
2. I totally agreed with the microsatellite instability and wanted to add in something about immunotherapy in the recurrent meningioma setting, so added the reference that the reviewer recommended and two additional ones.
3. I explained the positioning of the arrow by the neuropathologist who is part of the manuscript.
Soma Sengupta
Reviewer 2 Report
This is a well organized paper. The molecular pathogenesis of meningioma and vestibular schwannoma in NF2 is accurately described and analyzed. The graphic design is very helpful to provide a clear explanation of the paper
Author Response
As requested by reviewer 2, we ran spell-check and Grammar check in Microsoft Word, and I corrected the minor spell-check. We would like to thank reviewer 2 for their compliments.